# Acyclic Diene Metathesis (ADMET) Polymerization for the Synthesis of Chemically Recyclable Bio-Based Aliphatic Polyesters

**Kotohiro Nomura *** and **Xiuxiu Wang**

Department of Chemistry, Tokyo Metropolitan University, Hachioji, Tokyo 1920397, Japan
* Correspondence: ktnomura@tmu.ac.jp

**Abstract:** The recent developments of the synthesis of bio-based long-chain aliphatic polyesters by the acyclic diene metathesis (ADMET) polymerization of α,ω-dienes, derived from plant oils and bio-based chemicals, like bis(10-undecenoate) with isosorbide, using ruthenium-carbene catalysts are reviewed. The development of subsequent (one-pot) tandem hydrogenation produced saturated polyesters under mild conditions. The polymerizations under bulk (without solvent, 80–90 °C) or in ionic liquids (50 °C) under vacuum conditions enabled the synthesis of high molar mass polymers ($M_n$ > 30,000 g/mol). The polymerization performed by the molybdenum-alkylidene catalyst afforded the highest-molecular-weight polyesters (44,000–49,400 g/mol, in toluene at 25 °C) exhibiting promising tensile properties (strength and elongation at break) compared to polyethylene and polypropylene. Depolymerizations of these polyesters, including closed-loop chemical recycling, were also demonstrated. Catalyst developments (more active, under mild conditions) play a key role in the efficient synthesis of these materials.

**Keywords:** bio-based; polyester; metathesis polymerization; plant oil; circular economy; chemical recycling; tensile properties; homogeneous catalysts





## 1. Introduction

The development of sustainable polymers from renewable feedstocks attracts considerable attention from the viewpoints of the circular economy as well as green sustainable chemistry. Hydrocarbon-rich molecular biomasses, such as vegetable oils (castor, coconut, linseed, olive, palm, soybean, sunflower, etc.) presented as triglycerides with fatty acids, or fatty acid esters (FAEs) are naturally abundant and are recognized as low-cost molecular biomass products [1–11]. A study on bio-based advanced polyesters (exhibiting tunable mechanical properties and biodegradability), in particular, long-chain aliphatic polyesters (LCAPEs), are semicrystalline materials considered as a promising alternative of polyethylene [6,8]. The melting temperatures ($T_m$ values) of polyesters are generally influenced by the methylene length (and the direction of dipoles called the odd–even effect) [6,12–14]; the placement of longer methylene units should be effective for the obtainment of polyesters without softening them at elevated temperatures. It has been considered that the precise polymerization technique provides a new strategy and methodology for the design of macromolecular architectures.

Two condensation polymerization approaches—(i) condensation polymerization by transesterification (dicarboxylic acid and diol, etc.) and (ii) acyclic diene metathesis (ADMET) polymerization (nonconjugated α,ω-dienes)—and subsequent hydrogenation (Scheme 1) are considered for the synthesis from FAEs [6,8]. The ring-opening polymerization (ROP) approach from cyclic monomers can also be considered, but the method has a limited monomer scope; the method also faces the difficulty of catalysts enabling the synthesis of high molar mass polymers [15,16]. Studies on alternative approaches to polymers are also under investigation [17–20]. Moreover, the recent progress in the

development of olefin metathesis catalysts for the conversion of plant oils (FAEs) is well known [21–24].

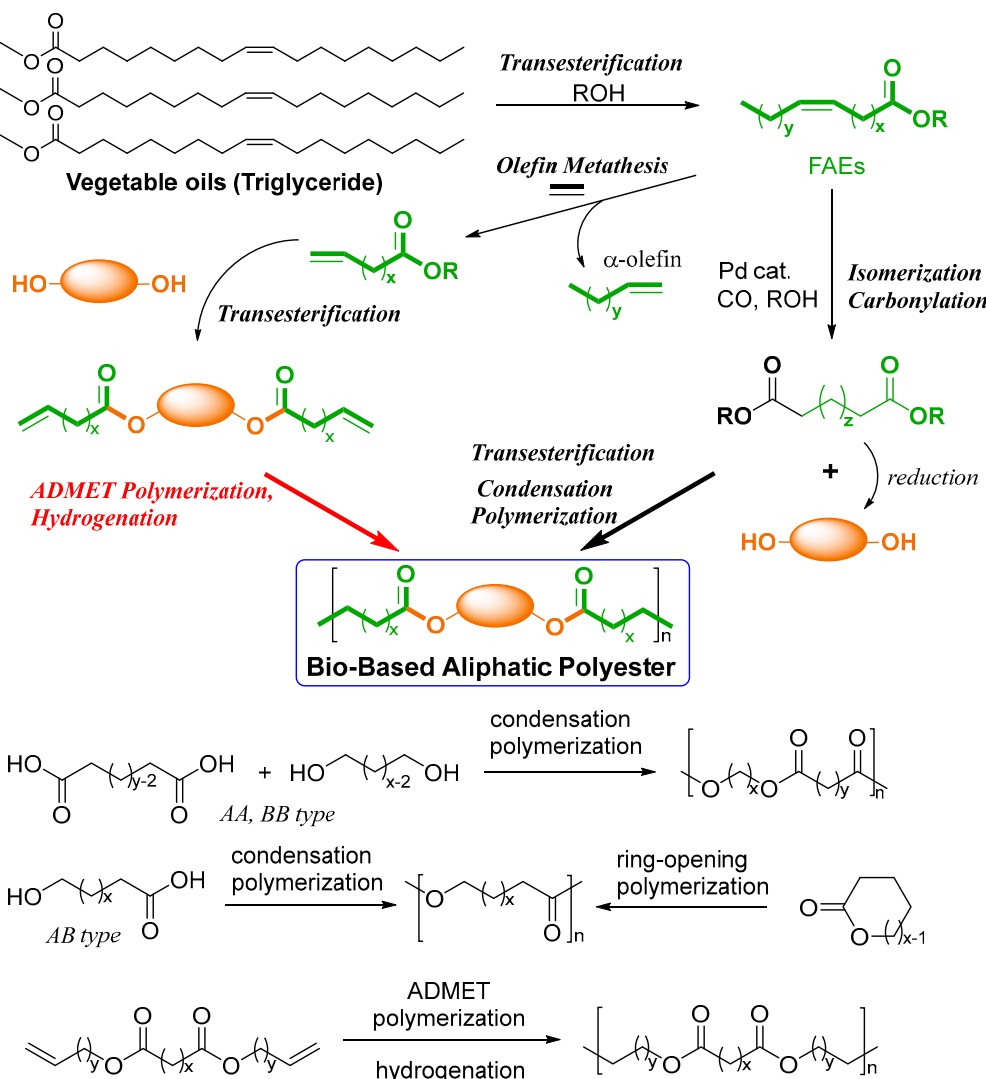

**Scheme 1.** Synthesis of bio-based polyesters from plant oil (triglycerides).

The conventional condensation polymerization approach through transesterification (ester bond exchange) requires high temperatures with the efficient removal of by-products (such as alcohols and water) to obtain high molar mass polymers with a high degree of polymerization ($DP_n$). For example, the synthesis of poly(ethylene terephthalate) from terephthalic acid (which must be purified) with excess ethylene glycol requires high temperatures up to 290 °C under a reduced pressure [25]. This method, however, seems to be difficult to apply for the synthesis of LCAPEs due to the difficulty of removing diols with high boiling points (e.g., 1,12-dodecanediol, 189 °C/12 Torr; 1,16-hexadecane diol 197–199 °C/3 mmHg). Moreover, precise stoichiometric control (hydroxy and carboxylic groups) is needed for this purpose [6,26–28]; polymerization with the precise stoichiometric ratios of diols (algae oil) and diesters ($C_{17}$ and $C_{19}$) is required to create high molar mass polymers ($M_n = 4.0 \times 10^4$) possessing a $T_m$ value of 99 °C [28].

The synthesis of bio-based aliphatic polyesters by adopting the ADMET polymerization [29–31] approach, especially using commercially available (Grubbs-type) ruthenium carbene catalysts, has been explored by many researchers, since the reactions do not require severe conditions for condensation polymerization through transesterification (de-

scribed above). In this mini review, reports concerning the synthesis of bio-based aliphatic polyesters by the ADMET approach are summarized.

## 2. Synthesis of Bio-Based Aliphatic Polyesters by ADMET Polymerization

### 2.1. Synthesis of Aliphatic Polyesters by ADMET Polymerization and Hydrogenation

Reports on the synthesis of bio-based polyesters by ADMET polymerization, especially using commercially available (Grubbs-type) ruthenium carbene catalysts $RuCl_2(PCy_3)_2(CHPh)$ (**G1**; Cy = cyclohexyl), $RuCl_2(PCy_3)(IMesH_2)(CHPh)$ (**G2**; IMesH$_2$ = 1,3-bis(2,4,6-trimethylphenyl) imidazolin-2-ylidene), and $RuCl_2(IMesH_2)(CH-2-O^iPr-C_6H_4)$ (**HG2**), shown in Scheme 2, are well known. These ruthenium catalysts have been employed [8] because these complexes can be readily available and do not require treatment with the strict Schlenk technique due to their insensitivities toward water and oxygen (better functional group tolerance) [32–35]. More recently, the example using a molybdenum-alkylidene catalyst (**Mo cat.**) [36–38], shown below, also demonstrates the synthesis of high molar mass polymers that exhibit good tensile properties [39].

Cy = cyclohexyl; Mes = 2,4,6-Me$_3$C$_6$H$_2$          R = OC(CH$_3$(CF$_3$)$_2$

**Scheme 2.** Ruthenium-carbene and molybdenum-alkylidene catalysts for the synthesis of aliphatic polyesters by ADMET polymerization.

The synthesis of a bio-based polyester, expressed as **PE1**, by the ADMET polymerization of undec-10-en-1-yl undec-10-enoate (**M1**), prepared by the reaction of 10-undecenoic acid with 10-undecenol (derived from castor oil) was reported by the group of Meier in 2008 [40]. The resultant **PE1** synthesized by **G2** (0.5 or 1.0 mol%, 80 °C, 24 h, Scheme 3) possessed a rather high molecular weight ($M_n$ = 22,000, 26,500), and the $M_n$ values were controlled by the addition of terminal olefins, such as methyl 10-undecenoate and stearyl acrylate [40]. In contrast, the group reported that the polymerization of bis(undec-10-enoate) with isosorbide (**M2**, Scheme 3) conducted at 70–100 °C under bulk conditions yielded rather low-molecular-weight polymers (**PE2**, Table 1) [41], whereas the $M_n$ values seemed to improve when the polymerizations were conducted at high temperatures and/or under nitrogen-purge conditions (for the removal of by-produced ethylene). This was probably due to the catalyst decomposition caused by conducting the reaction at 70–100 °C [42–47], because these ruthenium catalysts are known to decompose under these conditions to produce ruthenium-hydride species [44] and/or nanoparticles [46], which induce olefin isomerization and/or certain side reactions by the formed radicals [42–48]. **G2** showed a more significant degree of olefin isomerization compared to **G1** and a higher percentage of isomerization (estimated by GC-MS, after treating the mixture with MeOH-H$_2$SO$_4$ under reflux conditions) [41]. Later, the degree of isomerization was extensively suppressed when the polymerizations were conducted in the presence of benzoquinone [48].

The ADMET polymerization of **M1** by **G1** under high-vacuum conditions for two days produced **PE1** ($M_n$ = 28,000, $M_w/M_n$ = 1.9) and a subsequent hydrogenation step (Pd/C, 50 bar H$_2$, 60 °C) produced a saturated polyester (**HPE1**, PE-20.20, Scheme 4) [49]. The $T_m$ value (103 °C) achieved was somewhat low compared to the **HPE1** prepared by the condensation polymerization of 1,20-eicosanedioic acid with eicosane-1,20-diol ($T_m$ = 108 °C) to form 'regio-regular' ester groups, C(O)-O, aligned with the polymer chain (Scheme 4). It was thus suggested that the microstructural control directly affected the thermal property, as described above [6,14]. ADMET polymerizations of $\alpha,\omega$-dienes with different methylene chain lengths, di(icos-19-en-1-yl)tricosanedioate (**M3**) and di(tricos-22-

en-1-yl)tricosanedioate (**M4**), using **G1** and the subsequent olefin hydrogenation conducted by Ru(CHOEt)Cl$_2$(PCy)$_2$ (40 bar H$_2$, 70 °C, 2 d), prepared from **G1**, yielded the corresponding values of PE-38.23 (**HPE3**) and PE-44.23 (**HPE4**), respectively (Scheme 4) [50]. The polycondensation of 1,26-hexacosanedioate, prepared by the cross-metathesis of erucic acid, with the corresponding diol (produced by a reduction with LiAlH$_4$) with Ti(OBu)$_4$ also produced the corresponding polyester (**HPE5**, PE-26.26, $T_m$ = 114 °C) [51]. The thermal properties ($T_m$ values) of the resultant LCAPEs with different methylene lengths, prepared by ADMET [50] and polycondensation [51,52] approaches, revealed that the $T_m$ values achieved a constant value (Figure 1a) [50]. A linear relationship between the $T_m$ values and the number of ester groups in 1000 carbon atoms was observed (Figure 1b) [50]. Polyesters PE-26.26, PE-12.26 and PE-4.26 [51], and PE-18,18 [53] were also prepared by polycondensation.

**Scheme 3.** ADMET polymerization of castor oil-derived monomers (**M1**, **M2**) [40,41].

**Table 1.** Synthesis of **PE2** by ADMET polymerization using ruthenium catalysts [41] [1].

| Ru cat. | Temp. /°C | Nitrogen Purge [2] | $M_n$ [3] | $M_w/M_n$ [3] | Isomerization [4] /% |
|---|---|---|---|---|---|
| **G2** | 60 | no | 5600 | 1.65 | 48 |
| **G1** | 70 | no | 4400 | 1.57 | 3 |
| **G2** | 70 | no | 6000 | 1.71 | 49 |
| **G1** | 80 | no | 4750 | 1.56 | 4 |
| **G2** | 80 | no | 6100 | 1.61 | 69 |
| **G1** | 80 | yes | 6600 | 1.77 | 3 |
| **G2** | 80 | yes | 8400 | 1.75 | 76 |
| **G1** | 90 | no | 5450 | 1.69 | 3 |
| **G2** | 90 | no | 6200 | 1.65 | 66 |
| **G1** | 100 | no | 5000 | 1.61 | 42 [5] |

[1] Conditions: Ru cat 1.0 mol%, 5 h. [2] N$_2$ purge during polymerization. [3] GPC in THF vs. polystyrene stds. [4] Isomerized diesters (%) estimated with GC-MS after transesterification. [5] Unidentified side products.

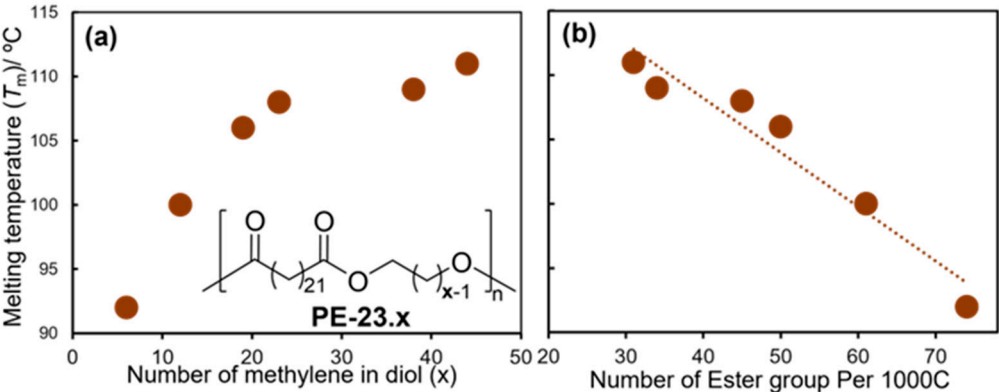

**Scheme 4.** Synthesis of linear polyesters (LCAPEs) [49–51].

**Figure 1.** Plots of melting temperature ($T_m$) vs. number of (**a**) methylene units (x) in diol and (**b**) ester groups per 1000 C (methylene) in PE-23.x [50].

The one-pot synthetic method used for the bio-based aliphatic polyesters by AD-MET polymerization and subsequent hydrogenation was demonstrated (Scheme 5) [54]. The polymerization of bis(undec-10-enoate)s with isosorbide (**M2**), isomannide (**M6**), 1,3-propanediol (**M7**), and 1,4-cyclohexanedimethanol (**M8**), derived from castor oil and glucose in chloroform by **G2** or **HG2** under a reduced pressure at 50 °C produced unsaturated polymers (expressed as **PE2** and **PE6**–**PE8**, respectively) [54]. The $M_n$ values in the produced polymers ($M_n$ = 11,900–15,900) were somewhat higher than those reported previously ($M_n$ = 4400–8400), conducted at 70–100 °C [41], and the $M_n$ values did not change, even under rather scaled-up conditions [54]. One reason for the obtainment of high-molecular-weight product could be that the degree of the catalyst decomposition was significantly suppressed by conducting the polymerization at 50 °C (and the polymerization was conducted under a continuously reduced pressure) [54].

As described above (Scheme 4) and below [55], conventional olefin hydrogenation requires a high hydrogen pressure and high temperature after the isolation of unsaturated polyesters after ADMET polymerization [49,50,55]. In contrast, one-pot hydrogenation under rather mild conditions (1.0 MPa, 50 °C, 3 h) was demonstrated following the addition of a small amount of Al$_2$O$_3$ (ca. 1 wt%) to the reaction mixture (Scheme 5). The completion of the olefin hydrogenation was confirmed by DSC thermograms (uniform compositions) due to the difficulty (accuracy of the integration of olefinic protons) of obtaining the

$^1$H NMR spectra. No significant differences in the $M_n$ and $M_w/M_n$ values were observed before/after hydrogenation [54].

**Scheme 5.** One-pot synthesis of bio-based polyesters by Ru-catalyzed ADMET polymerization and hydrogenation [46].

As shown in Figure 1b, the melting temperatures ($T_m$ values) of the polyesters are influenced by the methylene unit number (n). As shown in Scheme 6, the copolymerization of **M1** with undeca-1,10-diene (UDD) followed by olefin hydrogenation (H$_2$ 40 bar, 110 °C, 2 d) produced various LCAPEs with different chain lengths (ranging from 0.9 to 52.6 ester groups per 1000 carbon atoms), expressed as H$_2$-poly(**M1**-*co*-UDD) [55]. A linear correlation of the melting temperatures ($T_m$ values) with the average number of ester groups per methylene unit was thus demonstrated, whereas the ester group was incorporated in a random manner. A similar trend was observed in the copolymerization of **M2** with 1,9-decadiene (DD) and the subsequent one-pot hydrogenation [56]. The saturated polymers possessed $T_m$ values in the range of 71.7–107.6 °C, depending on the molar ratios of **M2** and DD.

**Scheme 6.** ADMET copolymerization of undec-10-en-1-yl undec-10-enoate (**M1**) or bis(undec-10-enoate) with isosorbide (**M2**) with nonconjugated dienes, and subsequent hydrogenation [55,56].

The polymerization of bis(undec-10-enoate)s with *D*-xylose (1,2-*O*-isopropylidene-$\alpha$-*D*-xylofuranose, **M9c**), and *D*-mannose (**M10**) by **G2** was studied under a dynamic-vacuum (0.1 mbar) condition without solvent (bulk) conditions (60–90 °C, 20 h, Scheme 7) [57]. The molecular weights of the resultant polymers (**PE9c**, **PE10**) were affected by the polymerization temperature employed and the monomer/Ru molar ratios. Conducting the polymerization at 90 °C under a low Ru concentration (0.1 mol%) seemed to be the optimized condition (**PE9c**: Ru, $M_n = 7.14$–$7.16 \times 10^4$, $M_w/M_n = 2.2$–$2.3$, **PE10**: $M_n = 3.24 \times 10^4$, $M_w/M_n = 2.4$) [49]. Due to the fact that the polymerization was conducted without a sol-

vent, the PDI ($M_w/M_n$) values were rather high due to the difficulty pf controlling the stirring [57]. Later, the polymerizations of *D*-xylose diester analogs with different methylene lengths (**M9**, x = 0, 2, 8, Scheme 7) and the corresponding diether analogs (**M11**) were explored [58]. The $M_n$ values of the resultant polymers decreased upon decreasing the methylene length, and the monomers did not possess a methylene spacer [58]. Some polymerization runs failed due to precipitation or the difficulty of performing isolations [58]. The resultant unsaturated polymers were amorphous, except **PE11a**, and both glass transition temperatures ($T_g$) increased after reducing the olefinic double bonds by treating them with *p*-toluenesulfonyl hydrazide as a reducing agent; most of the resultant saturated polymers (**HPE9** and **HPE11**) were amorphous, except **HPE9a** and **HPE11a** derived from the castor oil (10-undecenoate), suggesting that the placement of the methylene spacer was important (as shown in Figures 1a and 2) [58]. The resultant hydrogenated polymer films, especially the **HPE11a**-oriented film, exhibited a good tensile strength (43 MPa) with an elongation at a break of 155%; but, the hot-press film showed a much weaker tensile strength (7.8 MPa) with and improved elongation at the break (667%) [58].

**Scheme 7.** ADMET polymerization of α,ω-dienes containing *D*-xylose, *D*-mannose, vanillin, and eugenol as the monomer units [57–60].

The syntheses of polyesters containing vanillin (**PE12**) [59] afforded high-molecular-weight **PE12** ($M_n$ = 10,000, $M_w/M_n$ = 1.6) possessing a $T_g$ value of 4 °C (Scheme 7), whereas the polymerization of 4-allyl-2-methoxyphenyl 10-undecenoate (**M13**) by **G2** produced amorphous high molar mass polymers with low PDIs ($M_w/M_n$) with $T_g$ at −9.6 °C [60]. The ADMET polymerization of **M13** in the presence of 5-formylbenzene-1,2,3-triyl tris(undec-10-enoate) produced rather high molar mass network polymers [60].

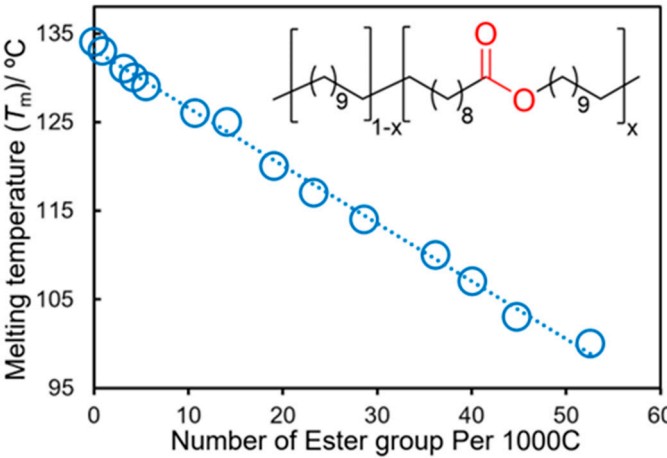

**Figure 2.** Plots of melting temperature ($T_m$) vs. number of ester groups per 1000 C (methylene units) in the hydrogenated copolymers, H$_2$-poly(**M1**-*co*-UDD)s [55].

The polymerization of trehalose bis(10-undecenoate) (**M14**) by **HG2** (4.0 mol%) in THF at 45 °C for 24 h (Scheme 8) produced semicrystalline polymers (**PE14**) possessing high molecular weights with unimodal molecular-weight distributions ($M_n$ = 13,200, $M_w/M_n$ = 2.1) with higher $T_m$ values (156 °C) [61]. Both the molecular weights and melting temperatures ($T_m$ values) of the resulting copolyesters with undec-10-en-1-yl undec-10-enoate (**M1**) decreased with the increase in the percentage of **M1** [61].

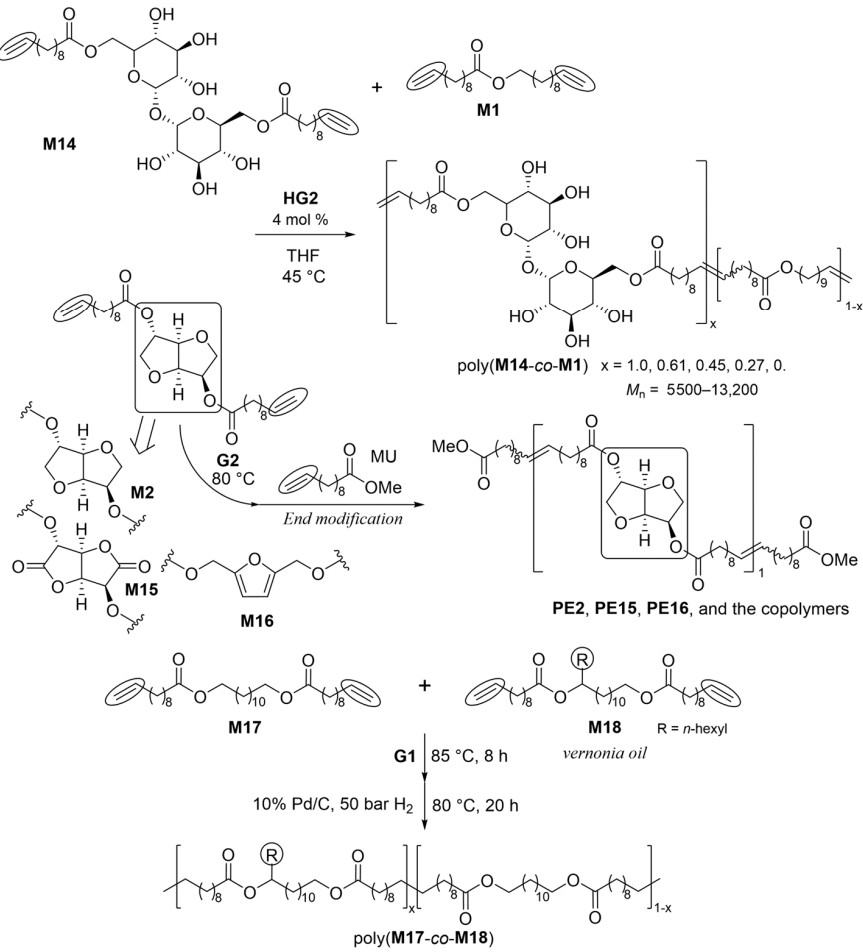

**Scheme 8.** Synthesis of bio-based copolyesters with different molar ratios [61–64].

The polymerization of bis(10-undecenoate)s with isosorbide (**M2**) and glucarodilactone (**M15**) and copolymerizations with different molar ratios were conducted in the presence of methyl-10-undecenoate (MU, 1.0 mol%) by using **G2** (1.0 mol%) at 80 °C for 16 h under a reduced pressure (Scheme 8) [62]. MU was employed as the monofunctional chain stopper (chain transfer reagent by the placement of the MU unit as the end group) [62]. The resultant polymers possessed high molecular weights with unimodal molecular-weight distributions. Copolymerizations with bis(hydroxymethylfuran) undecenoate (**M16**) were conducted [63]. The resultant **PE2** possessed a low $T_g$ value ($-10$ °C) compared to **PE15** ($T_g = 32$ °C), and the homopolymers, **PE2** and **PE15**, were brittle materials, whereas the copolyesters were rubbery materials possessing better tensile properties, an elastic behavior, as well as shape memory properties.

The copolymerizations of $\alpha,\omega$-dienes (linear **M17** and *n*-hexyl branched **M18**), derived from castor oil and vernonia oil, by **G1** at 85 °C, produced LCAPEs containing branches at a certain percentage (after subsequent hydrogenation by Pd/C, Scheme 8) [64]. These polymers were considered as LLDPE (linear low-density polyethylene) and VLDPEs (very low-density polyethylene) mimics. However, their DSC thermograms presented multiple melting temperatures, suggesting the compositions of the resultant copolymers were not uniform [64].

## 2.2. Synthesis of High Molecular-Weight Polymers Exhibiting Tensile Properties beyond Polyethylene and Polypropylene

There are many reports on the synthesis of bio-based aliphatic polyesters by the ADMET polymerization of $\alpha,\omega$-diene monomers containing carbohydrate units (such as **M2**, **M6**, **M9**, **M10**, **M14**, and **M15**) using ruthenium catalysts [41,54,57,58,61–63]; however, the reports on the synthesis of high-molecular-weight polymers (ca. $M_n \geq 30{,}000$ considered for their better mechanical properties, such as their films, shown below) are limited to date (Scheme 9) [58,62,63]. Catalyst decomposition was highly considered when metathesis polymerizations (reactions) were conducted at high temperatures (70–100 °C) and the subsequent isomerization and/or undesired side reaction caused by formed radicals were known [42–47]. The catalyst decomposition also caused the difficulty of separating the metal (present as ruthenium metal particles) from the resultant polymers, and this is often observed in metathesis polymerization chemistry, especially when using ruthenium catalysts. Moreover, the reported synthetic methods were conducted under direct-vacuum and bulk conditions without a solvent [58,62,63]; the methods thus presented the difficulty of stirring high-viscosity products [58] and was applicable to for the synthesis of amorphous or semicrystalline materials with $T_m$ values below 90 °C. Therefore, the development of the methods for the solution polymerization in the presence of appropriate solvent seemed to be better in terms of the process control (by lowering the viscosity of the reaction mixture under rather mild conditions to avoid catalyst decomposition) and the wide monomer scope.

ADMET polymerization is a condensation polymerization method that by-produces small molecules (ethylenes), and the removal is quite effective for the obtainment of high molar mass polymers under certain equilibrated conditions. Conducting the polymerization under continuous dynamic-vacuum and bulk conditions [58,62,63] is thus effective for this purpose. A consideration of these points shows that ionic liquids (ILs) can be considered as ideal solvents, not only due to their absent (or extremely low) vapor pressure and ability to provide homogeneous conditions due to their good miscibility with polymers, organic compounds, and metal catalysts, but also due to their high stability ranging from $-30$ to $>300$ °C [65–72]. Although olefin metathesis reactions in ILs are known, the reported examples for ADMET polymerizations are still limited [73–78].

More recently, the synthesis of high-molecular-weight polymers (**PE2**, $M_n = 32{,}200$–$39{,}200$) was demonstrated in the polymerization of $\alpha,\omega$-diene monomer (**M2**, dianhydro-*D*-glucityl bis(undec-10-enoate)) using the **HG2** catalyst in ionic liquids (ILs) under continuous-vacuum conditions at 50 °C (Scheme 10) [79]. The $M_n$ values were apparently higher than those reported previously ($M_n = 5600$–$14{,}700$) [41,54]. 1-*n*-Butyl-3-methyl imidazolium hexafluo-

rophosphate, [Bmim]PF$_6$, and 1-$n$-hexyl-3-methyl imidazolium bis(trifluoromethanesulfonyl) imide, [Hmim]TFSI, were found to be effective as solvents among a series of imidazolium salts and pyridinium salts. As summarized in Table 2, the method was also effective for the syntheses of high molar mass polymers containing isomannide (**PE6**), 1,4-cyclohexanedimethanol (**PE8**), and 1,4-butanediol (**PE7**) units as the diol segments used instead of isosorbide (**PE2**); the $M_n$ values did not decrease, even under the scale-up conditions (300 mg → 1.0 g scale) [79]. The tandem hydrogenation of the resultant unsaturated polymers (**PE2**) in the [Bmim]PF$_6$–toluene biphasic system upon the addition of Al$_2$O$_3$ (H$_2$ 1.0 MPa at 50 °C) produced the corresponding saturated polymers (**HPE2**).

$M_n$ = 56,000, $M_w/M_n$ = 1.8 $\qquad$ $M_n$ = 61,000, $M_w/M_n$ = 1.8

Conditions: **G2**, 80 °C, 16 h (under dynamic vaccum in bulk)

$M_n$ = 71,600, $M_w/M_n$ = 2.3 $\qquad$ $M_n$ = 60,800, $M_w/M_n$ = 3.5

Conditions: **G2**, 90 °C, 20 h (under dynamic vacuum in bulk)
(with mechanical over head stirrer)

**Scheme 9.** Selected reports for the synthesis of high-molecular-weight aliphatic polyesters by the acyclic diene metathesis (ADMET) polymerization of α,ω-diene monomers containing carbohydrate units [58,62,63].

**Scheme 10.** Synthesis of high-molecular-weight bio-based polyesters by ADMET polymerization in ionic liquids (ILs) and tandem hydrogenation, and depolymerization by olefin metathesis and transesterification [79].

**Table 2.** ADMET polymerizations of **M2**, **M6–M8** by **HG2** in [Hmim]TFSI [79] [1].

| Monomer | Yield [2]/% | $M_n$ [3] | $M_w/M_n$ [3] |
|---|---|---|---|
| **M2** | 93 | 39,200 | 1.95 |
| **M2**[4] | 86 | 37,500 | 1.91 |
| **M6** | 92 | 26,000 | 1.95 |
| **M7** | 89 | 33,400 | 2.30 |
| **M7** [4] | 87 | 34,900 | 1.82 |
| **M8** | 94 | 38,800 | 3.38 |

[1] Conditions: monomer (300 mg) in IL 0.14 mL (initial conc. 4.48 M (**M2**), 4.48 M (**M6**), 5.07 M (**M7**), 4.69 M (**M8**)), **HG2** 1.0 mol%, 50 °C in vacuo. [2] Isolated yield. [3] GPC data in THF versus polystyrene standards. [4] Reaction scale: monomer (1.0 g) in [Hmim]TFSI 0.30 mL (initial concentrations: 6.97 M (**M2**) and 7.80 M (**M7**)).

As described above, the polymerization of **M2** conducted in ILs with the continuous removal of by-produced ethylene afforded high molar mass polymers (Scheme 10) [79], whereas the polymerizations conducted in toluene or CHCl₃ (even under optimized conditions with the careful removal of ethylene) afforded polymers with $M_n$ values up to 15,000 [54]. The development of the method without using (expensive) ILs is favorable from a practical point of view.

We more recently demonstrated that the synthesis of higher molar-mass polymers ($M_n$ = 44,000–49,400 g/mol) could be achieved by polymerization in toluene using the molybdenum-alkylidene catalyst, Mo(CHCMe₂Ph)(2,6-Me₂C₆H₃)[OC(CH₃)(CF₃)₂] (**Mo cat.**, Scheme 11) [39]. As summarized by the results in Table 3, the $M_n$ values are affected by the **M2**/Mo molar ratios and amount of toluene used. As observed in the conventional ADMET polymerization, polymerization with low catalyst loading under high initial monomer conditions was suited to the condensation polymerization; it seemed that the $M_n$ value in **PE2** increased when the reaction scale was increased (90.5 (43.5 mg) → 261 μmol (543 mg)) with the increase in the initial monomer concentration (by varying the amount of toluene) [39]. This method is applicable to the other monomers (**M6**, **M19**). Olefinic double bonds in the resultant polymers were hydrogenated by using a rhodium catalyst under mild conditions (1.0 MPa, 50 °C), and no significant changes in the $M_n$ or PDI values of the polymers after hydrogenation were observed.

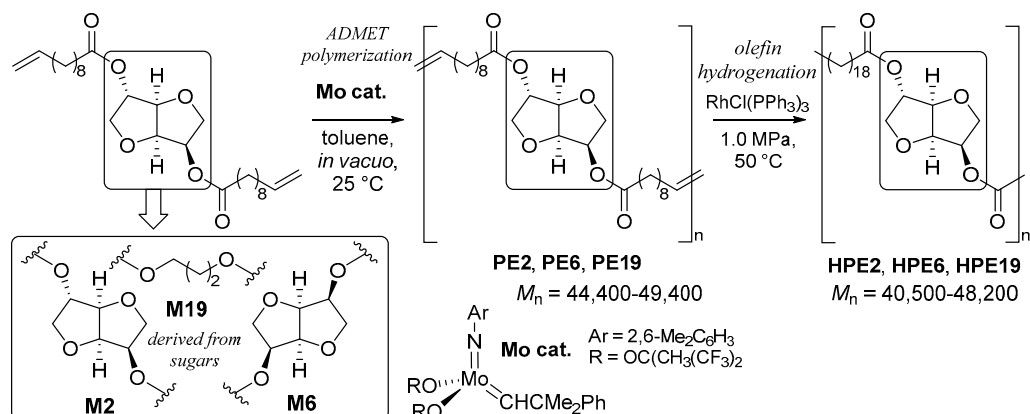

**Scheme 11.** Synthesis of high molecular mass polyesters by ADMET polymerization using the molybdenum catalyst [39].

It should be noted that both the tensile strength (stress) and elongation at break (strain) in the prepared polymer films of **HPE2** increased remarkably upon increasing the $M_n$ value (Figure 3) [39]; a fairly good linear correlation was observed between the stress and strain; and the **HPE2** sample with the highest $M_n$ value ($M_n$ = 48,200) exhibited a tensile strength of 39.7 MPa along with an elongation at break of 436%. The value was not only higher than PE-18,18, prepared with C₁₈ dimethyl dicarboxylate and the corresponding diol by condensation polymerization [9], but also poly(lactic acid) (PLA), poly(ethylene

terephthalate) (PET), high-density polyethylene (HDPE), low-density polyethylene (LDPE), and polypropylene (PP) [39,80]. The **PE2** sample before hydrogenation showed a higher strain (elongation at break) with less stress (tensile strength) compared to **HPE2**, and the isomannide-based **HPE6** showed a similar tensile property to the isosorbide-based **HPE2** [39]. The importance of the development of a synthetic method for the synthesis of high molar mass polymers by ADMET polymerization was thus demonstrated [39].

**Table 3.** ADMET polymerizations of **M2**, **M6**, and **M19** with the molybdenum catalyst (25 °C, 6 h) [39] [1].

| Monomer (µmol) | cat./mol% | Yield [2]/% | $M_n$ [3]/g·mol$^{-1}$ | $M_w/M_n$ [3] |
|---|---|---|---|---|
| **M2** (90.5) | 5.0 | 99 | 16,000 | 1.79 |
| **M2** (90.5) | 2.5 | 90 | 25,100 | 1.43 |
| **M2** (90.5) | 1.0 | 88 | 34,400 | 1.49 |
| **M2** (272) | 1.0 | 88 | 46,100 | 2.08 |
| **M2** (272) | 1.0 | 91 | 46,100 | 1.84 |
| **M6** (272) | 1.0 | 87 | 34,800 | 1.87 |
| **M19** (272) | 1.0 | 99 | 67,200 | 2.27 |
| **M2** (272) | 0.5 | 90 | 48,700 | 2.04 |
| **M2** (543) [4] | 0.5 | 91 | 49,400 | 2.47 |

[1] Conditions: Mo(CHCMe$_2$Ph)(N-2,6-Me$_2$C$_6$H$_3$)[OC(CH$_3$)(CF$_3$)$_2$]$_2$ (**Mo**), toluene (0.72 mL), quenched by C$_6$H$_5$CHO or 4-Me$_3$SiOC$_6$H$_3$CHO (for termination through Wittig-type cleavage). [2] Isolated yield (as MeOH insoluble fraction). [3] GPC data in THF (at 40 °C) vs. polystyrene standards. [4] Toluene: 1.0 mL.

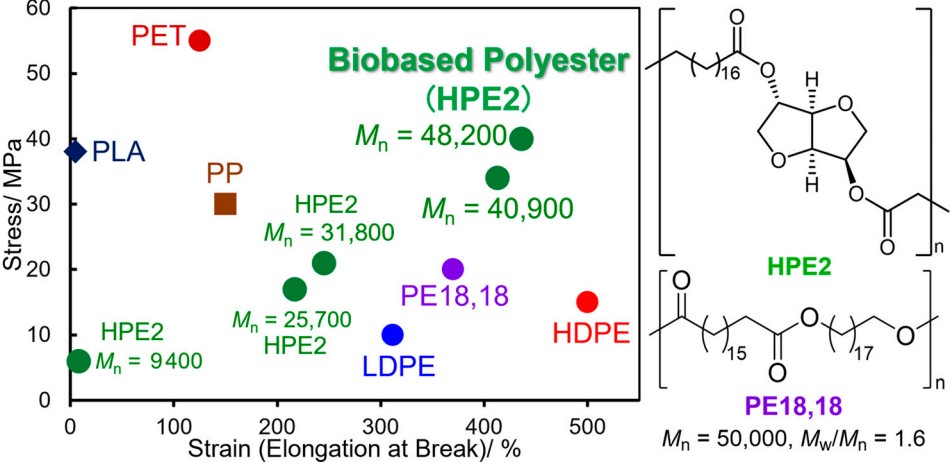

**Figure 3.** Plots of tensile (fracture) strengths and strains (elongations at breaks) of **HPE2** with different $M_n$ values. The plots of PE18.18 (polyester—18.18) [9], commercially available polyethylene terephthalate (PET), poly(lactic acid) (PLA), high-density polyethylene (HDPE), low-density polyethylene (LDPE), and polypropylene (PP) [39].

### 2.3. Chemical Recycling of Polyesters

PE18,18, prepared by the condensation polymerization of 1,18-octadeca dicarboxylic acid with 1,18-octadecanediol, was treated with MeOH (150 °C, 12 h) to produce a solid mixture consisting of dicarboxylic acid and diol after MeOH removal. The resultant solid was used for the subsequent condensation polymerization with Ti(O$^n$Bu)$_4$ to yield recycled PE18,18 with a high molecular weight ($M_n$ = 79,000, $M_w/M_n$ = 1.9, Scheme 12) [9]. Moreover, the treatment of polycarbonate (PC18, $M_n$ = 90,000, $M_w/M_n$ = 2.7), prepared by the condensation polymerization of 1,18-octadecane diol with diethyl carbonate (DEC) in the presence of LiH, with a 10 wt% KOH ethanol solution (at 120 °C, 24 h) exclusively produced 1,18-octadecanediol (yield: 98%, and purity: 99% after recrystallization from MeOH). The subsequent polycondensation with DEC produced recycled PC18 without a reduction in the $M_n$ value ($M_n$ = 70,000, $M_w/M_n$ = 3.4), which exhibited similar properties as the fresh sample [9]. These results indicate the possibility of closed-loop chemical recycling.

Although the conventional method for the depolymerization of polyester requires excess acid or base materials (or MeOH under high temperatures), more recently, exclusive acid- and base-free chemical conversions of polyesters (poly(ethylene adipate) (PEA), poly(butylene adipate) (PBA), poly(ethylene terephthalate) (PET), and poly(butylene terephthalate) (PBT)) into the corresponding monomers (diethyl adipate, diethyl terephthalate, ethylene glycol, and 1,4-butane diol) by transesterification with ethanol using the Cp'TiCl$_3$ (Cp' = Cp, Cp*) catalyst were demonstrated [81,82]. The depolymerizations proceeded the completed conversions (>99%) of PET and PBT to afford diethyl terephthalate and ethylene glycol or 1,4-butanediol exclusively (selectivity > 99%, 150–170 °C, Ti 1.0 or 2.0 mol%) [82]. The resultant reaction mixture after the depolymerization of PBA with ethanol by the CpTiCl$_3$ catalyst (1.0 mol%, 150 °C, 3 h), consisting of diethyl adipate and 1,4-butanediol, was heated at 150 °C in vacuo for 24 h to afford high-molecular-weight recycled PBA with a unimodal molecular-weight distribution ($M_n$ = 11,800, $M_w/M_n$ = 1.6, Scheme 12), demonstrating the possibility of one-pot (acid and base free) closed-loop chemical recycling [82]. The method can also be applicable to bio-based aliphatic polyesters; the reaction of **HPE2** with ethanol by CpTiCl$_3$ afforded the corresponding dicarboxylic acid and isosorbide products exclusively [79]. Since the depolymerization (transesterification) method by titanium catalysts can be applied to various polyesters, including bio-based ones [79,81,82], the importance of the basic concept of the one-pot method can be emphasized.

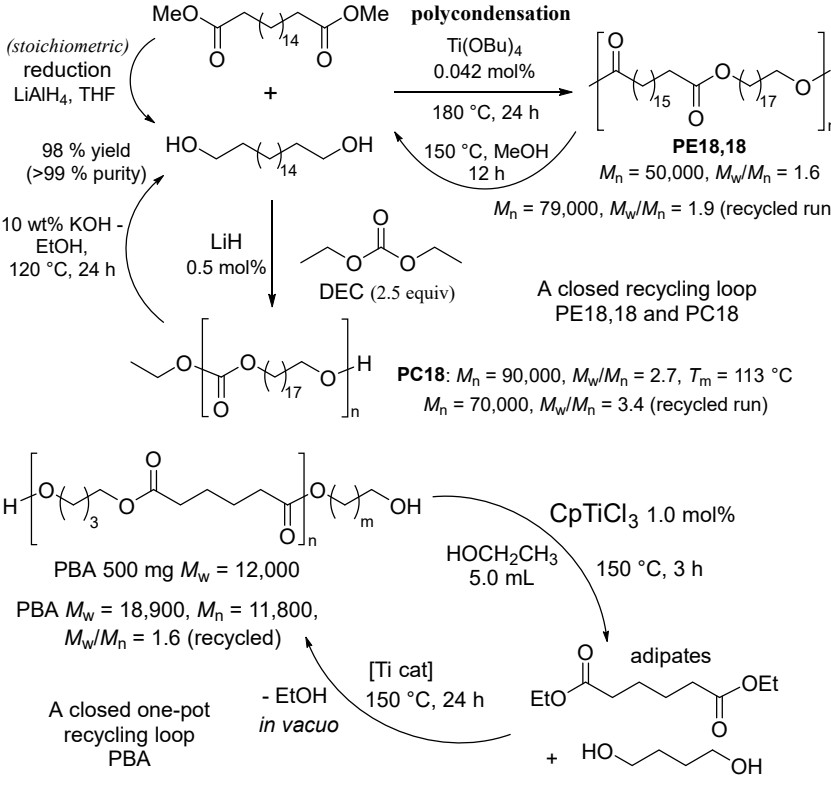

**Scheme 12.** Closed-loop chemical recycling of polyesters [9,82].

### 3. Concluding Remarks

This review summarizes the recent developments for the synthesis of bio-based LCAPEs by the acyclic diene metathesis (ADMET) polymerization of α,ω-dienes, derived from plant oils and bio-based chemicals (carbohydrates and their derivatives) in the presence of ruthenium-carbene catalysts (**G1**, **G2**, **HG2**; Scheme 2). The development of subsequent (one-pot) tandem hydrogenation afforded saturated polyesters under mild conditions. Reported examples for the synthesis of high-molecular-weight polymers are still limited; polymerizations under bulk conditions (without solvent, 80–90 °C) or in ionic liquids (50 °C) under vacuum conditions enabled the synthesis of high molar mass

polymers ($M_n > 30,000$) that exhibited better mechanical properties than films. However, high-temperature polymerizations (at 70–100 °C) created the possibility of catalyst decompositions. The polymerizations using the molybdenum-alkylidene catalyst afforded the highest molecular-weight polyesters (44,000–49,400 g/mol) even in toluene at 25 °C. Hydrogenated polyester films, prepared by the polymerization of bis(10-undecenoate) with isosorbide and the subsequent hydrogenation phase, exhibited promising tensile properties (strength and elongation at break) beyond polyethylene and polypropylene. The significant effects of molecular weight on the tensile properties were demonstrated, clearly indicating the importance of the synthesis of high molar mass polymers to produce better materials properties. The reported procedures for the closed-loop chemical recycling of polyesters by depolymerization and re-polymerization methods were also introduced. The depolymerization of poly(butylene adipate) (PBA) with ethanol using the CpTiCl$_3$ catalyst afforded diethyl adipate and 1,4-butandiol exclusively, and the subsequent polycondensation produced PBA without a loss of the $M_n$ value. Catalyst developments (more active, under mild conditions) play a key role in efficient synthesis practices.

**Author Contributions:** Conceptualization, project administration, funding acquisition: K.N.; writing—original draft preparation: X.W. and K.N.; writing—review and editing: K.N. All authors have read and agreed to the published version of the manuscript.

**Funding:** This project was partly supported by (Grant Number: JPMJCR21L5), JST SICORP (Grant Number: JPMJSC19E2), Japan, and the Tokyo Metropolitan Government Advanced Research (Grant Number: R2-1). XW thanks the Tokyo Metropolitan government (Tokyo Human Resources Fund for City Diplomacy) for the pre-doctoral fellowship.

**Data Availability Statement:** No new data were created or analyzed in this study. Data sharing is not applicable to this article.

**Acknowledgments:** K.N. would like to express his heartfelt thanks to the laboratory members who contributed as co-authors in the cited references for their wonderful contributions.

**Conflicts of Interest:** The authors declare no conflicts of interest.

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
