# Peer review of "Acyclic Diene Metathesis (ADMET) Polymerization for the Synthesis of Chemically Recyclable Bio-Based Aliphatic Polyesters"

_catalysts, doi:10.3390/catal14020097_

Round 1
Reviewer 1 Report
Comments and Suggestions for Authors
The review 'Acyclic Diene Metathesis (ADMET) Polymerization for Synthesis of Chemically Recyclable Bio-based Aliphatic Polyesters' presents and discusses the current state and prospects of the use of ADMET for the development of oleochemical polymer materials. The manuscript contains detailed and exhaustive analysis of the actual references. The review will be interesting for a general audience, the quality of presentation meets requirements of the Catalysts journal, and I recommend to accept this work for publication.
Please pay attention to some recommendations and techical remarks below to improve the quality of presentation.
1. The Scheme 1 has duplicate content: the topic of the review is the use of ADMET polymerization, and I recommend the use of colors for this approach, with presentation of alternative routes (transesterification/condensation and ROP) in black.
2. Line 58 – Grubbs catalysts per se are not effective in oleochemistry, it should be noted that further Ru-based catalyst generations were designed and adopted, with appropriate references.
3. Alternative approaches to polymers, based on oleochemicals, should be mentioned in this review:
ACS Sustainable Chem. Eng. 8 (2020) 10633. doi:10.1021/acssuschemeng.0c03432
Angew. Chem. Int. Ed. 56 (2017) 7589. doi:10.1002/anie.201702796
Green Chem. 23 (2021) 4255. doi:10.1039/d1gc00955a
ChemSusChem 4 (2011) 1052. doi:10.1002/cssc.201100187
4. In the Section 2.3, chemical recycling of PBA and PBAT is discussed. These copolymers are not relevant to the theme of the review, their mention has to be justified, more detailed discussion of the recycling of bio-based polyesters is needed in this Section. The mention of PBA in the Conclusion did not seem appropriate.
Technical remarks:
Line 46 – repl. 'with efficient removals of by product' by 'with efficient removal of by-product'
Line 47 – repl. 'obtainment' by 'obtaining'
Line 48 – remove PET, this abbreviation is not used further
Line 52 and below – repl. 'mmHg' by 'Torr'
Line 54 – repl. 'should be required' by 'is needed'
Line 67 – 'There have been' – unsuccessful turn
Line 82 – repl. 'prepared by' by 'prepared from'
Line 84 – repl. 'prepared by G2' by 'synthesized using G2'.
Line 88 and below – please distinguish between hyphen (-) and dash (–) throughout the text
Line 121 and below – please use degree symbol, °, throughout the text.
Line 132 – permission in needed
Line 172 – permission in needed
Line 176 – repl. 'in the resultant' by 'of the resultant'
Line 182 and below – repl. 'analogues' by 'analogs'
Line 331 – the quality of the Figure 3 is inappropriate, permission in needed
Comments on the Quality of English Language
In places there are unfortunate phrases and sentences, definite article is often missed. The text needs editing.
Author Response
Dear Reviewer 1:
Thank you so much for your helpful comments and we have revised the manuscript according to your comments. The point-by point responses are as follows.
- The Scheme 1 has duplicate content: the topic of the review is the use of ADMET polymerization, and I recommend the use of colors for this approach, with presentation of alternative routes (transesterification/condensation and ROP) in black.
Thank you for your kind comment and we have revise the Scheme 1 according to your comment.
- Line 58 – Grubbs catalysts are not effective in oleochemistry, it should be noted that further Ru-based catalyst generations were designed and adopted, with appropriate references.
Thank you for your comment, and we have added several references concerning recent catalysts for ethenolysis of plant oils. I suppose this could be fine for your request.
- Alternative approaches to polymers, based on oleochemicals, should be mentioned in this review:
ACS Sustainable Chem. Eng. 8 (2020) 10633. doi:10.1021/acssuschemeng.0c03432
Angew. Chem. Int. Ed. 56 (2017) 7589. doi:10.1002/anie.201702796
Green Chem. 23 (2021) 4255. doi:10.1039/d1gc00955a
ChemSusChem 4 (2011) 1052. doi:10.1002/cssc.201100187
Thank you for your comment, and we have added the references in the revised manuscript. I believe that this could be fine.
- In the Section 2.3, chemical recycling of PBA and PBT is discussed. These copolymers are not relevant to the theme of the review, their mention has to be justified, more detailed discussion of the recycling of bio-based polyesters is needed in this Section. The mention of PBA in the Conclusion did not seem appropriate.
Thank you for your critical comment. We have introduced the results in chemical recycling of biobased polymers and the fact shown in section 3 can be applied to conventional polyesters. I have added more description for better explanation. I highly hope that this could fulfill your request.
Technical remarks:
Line 46 – repl. 'with efficient removals of by product' by 'with efficient removal of by-product'
Line 47 – repl. 'obtainment' by 'obtaining'
Line 48 – remove PET, this abbreviation is not used further
Line 52 and below – repl. 'mmHg' by 'Torr'
Line 54 – repl. 'should be required' by 'is needed'
Line 67 – 'There have been' – unsuccessful turn
Line 82 – repl. 'prepared by' by 'prepared from'
Line 84 – repl. 'prepared by G2' by 'synthesized using G2'.
Line 88 and below – please distinguish between hyphen (-) and dash (–) throughout the text
Line 121 and below – please use degree symbol, °, throughout the text.
Line 132 – permission in needed
Line 172 – permission in needed
Line 176 – repl. 'in the resultant' by 'of the resultant'
Line 182 and below – repl. 'analogues' by 'analogs'
Line 331 – the quality of the Figure 3 is inappropriate, permission in needed
Thank you for your comments and we have revised (including revision of Figures 1-3).
Reviewer 2 Report
Comments and Suggestions for Authors
This manuscript described progress on the synthesis of biobased aliphatic polyesters via acyclic diene metathesis polymerization approach. The topic is much interesting and the oranization is quite well. Therefore, I would like to recommend this paper acception for the publication.
Author Response
Dear reviewer 2:
Thank you for your comments. We appreciate your reviewing.
Best regards,
Kotohiro Nomura
Reviewer 3 Report
Comments and Suggestions for Authors
The manuscript provides a comprehensive review of recent developments in the synthesis of bio-based long-chain aliphatic polyesters (LCAPEs). These polyesters are synthesized using acyclic diene metathesis (ADMET) polymerization of αω-dienes derived from plant oils and bio-based chemicals like bis(10-undecenoate) with isosorbide. The use of ruthenium-carbene catalysts and the development of subsequent one-pot tandem hydrogenation processes are highlighted. These advancements have enabled the synthesis of high molar mass polymers (Mn > 30000 g/mol) with promising tensile properties, surpassing those of polyethylene and polypropylene​​.
One of the key aspects of this research is the use of ruthenium-carbene catalysts, specifically Grubbs type catalysts, and the more recent molybdenum-alkylidene catalysts. These catalysts facilitate the ADMET polymerization process and contribute to the synthesis of high molar mass polymers with good tensile properties​​. The manuscript also details a one-pot synthetic method for bio-based aliphatic polyesters, which combines ADMET polymerization and hydrogenation. This method, conducted under reduced pressure at lower temperatures (50 ºC), yields unsaturated polymers with higher molecular weights than those produced at higher temperatures (70-100 ºC). This increase in molecular weight is attributed to the suppression of catalyst decomposition at lower temperatures​​. Furthermore, the review discusses the limitations encountered in the synthesis of high molecular weight polymers, such as catalyst decomposition at higher temperatures and challenges in processing due to high viscosity. These issues highlight the need for developing methods that allow for better process control and a wider range of monomers. Ionic liquids (ILs) are proposed as ideal solvents for ADMET polymerization due to their stability and miscibility with polymers and catalysts​​. Importantly, the review emphasizes the significance of molecular weight in determining the tensile properties of the polymers. Higher molecular weight polymers exhibit improved tensile strength and elongation at break. These properties are crucial for the development of superior materials, indicating the importance of synthesizing high molar mass polymers through ADMET polymerization​​. Lastly, the manuscript discusses the potential for closed-loop chemical recycling of polyesters. This process involves the depolymerization and re-polymerization of polyesters, offering a sustainable approach to polymer production and waste management​​.
In conclusion, this review comprehensively covers the advancements in the synthesis of bio-based LCAPEs through ADMET polymerization, highlighting the importance of catalyst development, polymerization conditions, and molecular weight on the properties of the resulting polymers. The potential for chemical recycling of these materials is also a significant aspect of this research, contributing to the sustainability of polymer production.
Based on the current manuscript, it is recommended that this manuscript be published in Catalyst.
Author Response
Dear Reviewer 3:
Thank you so much for your kind reviewing. We appreciate your kind support.
Best regards,
Kotohiro Nomura